# Pain Management at the End of Life in the Emergency Department: A Narrative Review of the Literature and a Practical Clinical Approach

**DOI:** 10.3390/jcm12134357

**Published:** 2023-06-28

**Authors:** Sossio Serra, Michele Domenico Spampinato, Alessandro Riccardi, Mario Guarino, Andrea Fabbri, Luciano Orsi, Fabio De Iaco

**Affiliations:** 1Emergency Department, Maurizio Bufalini Hospital, 47521 Cesena, Italy; sossio.serra@gmail.com; 2Department of Translational Medicine and for Romagna, University of Ferrara, 44124 Ferrara, Italy; 3Pronto Soccorso Ospedale di Imperia, 18100 Imperia, Italy; dottriccardi@gmail.com; 4UOC MEU Ospedale CTO-AORN dei Colli Napoli, 80131 Napoli, Italy; marioguarino63@gmail.com; 5Emergency Department, AUSL Romagna, Presidio Ospedaliero Morgagni-Pierantoni, 47121 Forlì, Italy; andrea.fabbri@auslromagna.it; 6Palliative Care Physician and Scientific Director of “Rivista Italiane di Cure Palliative”, 26013 Crema, Italy; orsiluciano@gmail.com; 7Struttura Complessa di Medicina di Emergenza Urgenza Ospedale Maria Vittoria, ASL Città di Torino, 10144 Torino, Italy; fabiodeiaco@aslcittaditorino.it

**Keywords:** pain management, emergency department, end of life care, palliative care, acute pain, chronic pain, total pain, palliative need evaluation, pain evaluation, analgesic drugs

## Abstract

Access to pain management is a fundamental human right for all people, including those who are at the end of life (EOL). In end-stage patients, severe and uncontrolled pain is a common cause of admission to the emergency department (ED), and its treatment is challenging due to its complex, often multifactorial genesis. The aim of this narrative review was to identify the available literature on the management of severe EOL pain in the ED. The MEDLINE, SCOPUS, EMBASE, and CENTRAL databases were searched from inception to 1 April 2023 including randomised controlled trials, observational studies, systemic or narrative reviews, case reports, and guidelines on the management of EOL pain in the ED. A total of 532 articles were identified, and 9 articles were included (5 narrative reviews, 2 retrospective studies, and 2 prospective studies). Included studies were heterogeneous on the scales used and recommended for pain assessment and the recommended treatments. No study provided evidence for a better approach for EOL patients with pain in the ED. We provide a narrative summary of the findings and a review of the management of EOL pain in clinical practice, including (i) the identification of the EOL patients and unmet palliative care needs, (ii) a multidimensional, patient-centred assessment of the type and severity of pain, (iii) a multidisciplinary approach to the management of end-of-life pain, including an overview of non-pharmacological and pharmacological techniques; and (iv) the management of special situations, including rapid acute deterioration of chronic pain, breakthrough pain, and sedative palliation.

## 1. Introduction

Pain is a common symptom at the end of life and can manifest itself in both acute and chronic forms. In particular, severe pain at the end of life is common, with a high prevalence two years before death and a marked increase in the last four months [1]. Hagarty et al. found that 17% of the more than 20,000 participants in their study reported severe pain daily. No association was found between the cause of impending death or the medical setting and the occurrence of pain at the end of life. [2]. Seventy-five to ninety percent of patients with an advanced disease require opioid therapy for severe chronic pain [3], but cancer patients are not the only ones who experience end-of-life pain. In a systematic review, 66% of patients with moderate to severe COPD and 21% to 77% of patients with end-stage lung disease reported pain [4]. The prevalence of moderate to severe pain ranges from 37% to 50% in patients with end-stage renal disease [5], while the prevalence in patients with end-stage liver disease ranges from 30% to 79% [6]. In any stage of heart failure [7], AIDS [8] or neurological degenerative diseases [9,10], the prevalence of pain is high, showing that it is a significant issue.

Regardless of the clinical setting or cause, the management of severe pain is a highly complex problem. According to the revised 2020 definition of the International Association for the Study of Pain, pain is “an unpleasant sensory and emotional experience associated with or resembling that associated with actual or potential tissue damage” [11]. Chronic pain, on the other hand, should be considered a health condition in its own right [12], and its treatment should be based on a comprehensive biopsychosocial model that recognises the complex, multidimensional nature of the condition and examines the pathophysiology of pain in relation to a variety of cognitive, affective, behavioural, and social characteristics [13]. Years of suffering and fear of death can profoundly alter the perception and meaning of pain [14,15,16,17], leading to “total pain” in terminally ill patients [18]. This term was coined by Dame Cicely Saunders, founder of the hospice movement, who proposed that pain should be understood as having physical, psychological, social, emotional, and spiritual components. The ICD-11 [19] classifies chronic pain (defined as pain lasting at least three months) into seven categories: chronic primary pain, chronic cancer pain, chronic post-traumatic pain, chronic neuropathic pain, head and facial pain, visceral pain, and musculoskeletal pain. Moreover, pain can be classified as somatic, visceral, or neuropathic, depending on its origin and the type of fibres involved in transmitting pain signals to the brain (A-beta, A-delta, and C fibres). In addition, some types of interventions such as transfers, suctioning of secretions, and dressings for skin ulcers, can cause acute end-of-life pain. All types of pain can occur at the end of life in both acute and chronic forms. Careful pain assessment is essential to identify the specific pain characteristics and guide treatment.

End-stage patients are often admitted to the emergency department because of severe, uncontrolled pain. Admission to the ED often occurs when symptoms are unbearable and cannot be adequately treated at home to intensify palliative care [20,21]. Nevertheless, pain is often multifactorial, and only one of many symptoms’ patients may present with at the end of life, making management decisions difficult [17,18,22,23].

The aim of this manuscript was to (i) identify the available literature on the management of severe EOL pain in the emergency department (ED) and (ii) provide a clinical review on the identification of patients worthy of palliative care and on the multidimensional approach to the identification and management of pain in EOL patients in the ED, taking into account the available literature and the experience of the SedoAnalgesia Group (SAU) of the Società Italiana di Medicina d’Emergenza-Urgenza (SIMEU).

## 2. Materials and Methods

This review followed the Preferred Reporting Items for Systematic Reviews and Meta-Analyses (PRISMA) 2021 version and the Preferred Reporting Items for Systematic Reviews and Meta-Analyses extension for Scoping Reviews (PRISMA-ScR) guidelines (Appendix B) [24]. No ethical approval was required for this study. This narrative review is registered on the Open Science Framework (OSF), DOI 10.17605/OSF.IO/TGZ7K. 

### 2.1. Eligibility Requirements

Randomised clinical trials (RCTs), prospective or retrospective observational studies, case reports, surveys, systematic reviews, narrative reviews, guidelines and position papers on pain management (by pharmacological or non-pharmacological treatments) in terminally ill patients over 18 years of age in the emergency department were included. Studies on topics other than pain management in EOL patients in the ED, studies conducted outside the ED, unavailable full-text studies, duplicate studies, studies not in English, and studies on populations other than humans were excluded.

### 2.2. Source of Information and Search Strategy

A comprehensive review was conducted by 20 March 2023. The databases MEDLINE, SCOPUS, EMBASE and “Cochrane Central Register of Controlled Trials” (CENTRAL) were searched. The search strategy of each database was modelled using the following medical subject headings (MeSH) and Boolean operators: “emergency department” AND (“pain management” OR “analgesia”) AND (“terminally ill” OR “palliative care” OR “actively dying” OR “end of life”). Two independent reviewers (SS and SMD) assessed relevant manuscripts from each database for possible inclusion. Disagreements about eligibility were resolved by a third reviewer (FDI). The full texts were reviewed by two reviewers (MG and AR), and discrepancies were resolved with the help of a third author (AF).

### 2.3. Extraction of Data

The following information has been extracted (if applicable): author, publication year, manuscript type, primary outcome or purpose of the study, population, sample size, age of included patients, pain scale used, analgesics assessed with dosage, and author’s conclusion.

### 2.4. Summary of the Findings

Data were presented with an overview of the number, type, and distribution of included studies, which was followed by a narrative synthesis and presentation of results. A corresponding table with a synthesis of the results is provided, and each study is briefly discussed. No quantitative data meta-analysis was conducted.

### 2.5. Definitions Used

Due to the varying and inconsistent definitions of terms in the literature, this review employs the terms outlined in the 2015 Australian Commission on Safety and Quality in Health Care National Consensus Statement [25]:

Actively dying: the hours, days, and sometimes weeks preceding a patient’s death. This is occasionally called the “terminal phase”.

End-of-life: The period during which a patient is suffering from a terminal illness, even if the illness’s prognosis is uncertain or unknown. This includes people with (i) advanced, progressive, incurable conditions; (ii) general frailty and concomitant conditions that make death within 12 months likely; (iii) pre-existing conditions that put them at risk of dying from a sudden acute crisis in their condition; and (iv) life-threatening acute conditions caused by sudden catastrophic events. 

End-of-life care: consists of physical, mental, and psychosocial evaluations, care, and treatment administered by medical professionals and support staff. It also includes support for families and carers as well as care for the deceased patient’s body. 

Palliative care: a treatment approach that enhances the quality of life for patients and their families facing a life-limiting illness by preventing and alleviating suffering. This includes the early detection, accurate evaluation, and treatment of pain and other (physical, psychosocial, and spiritual) problems. 

The terminal phase: the hours, days, and sometimes weeks preceding a patient’s death. This is sometimes referred to as the patient’s “actively dying” state.

The World Health Organisation defines palliative care as “an essential component of person-centred, integrated health services”. Alleviating severe health-related physical, psychological, social and spiritual suffering is a global ethical responsibility. For patients with cardiovascular disease, cancer, severe organ failure, drug-resistant tuberculosis, severe burns, end-stage chronic disease, acute trauma, extreme prematurity or extreme frailty in old age, palliative care may be required and must be available at all levels of care [26]. Often, the terms “active dying”, “end of life”, “end of life” and “dying” are used interchangeably to identify patients who are eligible for hospice. Dr Cicely Saunders founded hospice care in the 1960s to provide comprehensive palliative care for terminally ill patients. In some healthcare organisations, patients are eligible for hospice care if their prognosis is less than six months and they or their family agree with the hospice philosophy of comfort care in relation to their terminal illness [27]. In early palliative care, curative treatments are given at the same time as palliative care; in end-of-life palliative care, palliative treatments are given after curative treatments have been discontinued.

The pain terms and definitions adopted in this narrative review are according to the International Association for the Study of Pain, and they are reported in Table 1 [28].

## 3. Results

### 3.1. Result of the Database Search

A total of 532 articles were identified (142 in MEDLINE, 17 in SCOPUS, 303 in EMBASE and 70 in CENTRAL), and no article was excluded before screening. After reviewing the titles, abstracts and full texts, 212 articles were excluded because they led to another topic, 310 were excluded because they addressed another topic about EOL in ED, 126 were excluded because they addressed another topic about EOL patients outside ED, 12 were excluded because the full text was not available, 8 were excluded because they were not in English, and no article was excluded because it was not conducted in humans. In total, 19 articles were eligible for inclusion, and after removing duplicates, 9 articles were included (Bell, 2018 [29], Chan, 2021 [30], Coine, 2021 [31], de Oliveira, 2021 [32], Lamba, 2010 [33], Long, 2020 [34], Rojas, 2016 [35], Siegel, 2017 [36], Solberg, 2015 [37]) (see Figure 1 for the PRISMA flow diagram for the studies selection process). 

### 3.2. General Characteristics of the Included Studies

One study was a prospective observational study (Coyne, 2021 [31]), three were retrospective studies (Chan, 2021 [30], De Oliveira, 2021 [32], Rojas, 2016 [35]) and five were narrative reviews (Bell, 2018 [29], Lamba, 2010 [33], Long, 2020 [34], Siegel, 2017 [36], Solberg, 2015 [37]). One study examined the management of geriatric patients at the end of life (Bell, 2018 [29]), two studies examined oncology patients in palliative care (Coyne, 2021 [31] and Long, 2020 [34]), two studies examined patients in hospice or in need of hospice care (Lamba, 2010 [33], Siegel, 2017 [36]), and three studies examined end-of-life patients who needed palliative care for any reason (de Oliveira, 2021 [32], Rojas, 2016 [35], Solberg, 2015 [37]). Regarding age, one narrative review (Bell, 2018 [29]) and four observational studies (Chan, 2021 [30], Coyne, 2021 [31], de Oliveira, 2021 [32], Rojes, 2016 [35]) focused on adult patients, while three narrative reviews did not include information on the age of the patients studied. See Table 2 for the general characteristics of the included studies.

#### 3.2.1. Pain Assessment

Four studies recommended the use of the numerical rating scale, the Wong–Baker FACES scale, the verbal rating scale, or the Pain Assessment in Advanced Dementia scale, while four studies made no recommendation. 

#### 3.2.2. Drugs Prescription

Observational studies reported the following: one retrospective study reported the use of opioids in 70.2% of patients (Chan, 2021 [31]); two studies reported the use of NSAIDs, paracetamol, tramadol, and long- and short-acting opioids (Coyne, 2021 [31], de Oliveira, 2021 [32]); and two studies reported the use of only morphine via an intravenous bolus and drip (Lamba, 2010 [33], Rojas, 2016 [35]). The included reviews recommended the use of paracetamol, morphine or hydromorphone (Bell, 2018 [29]); sustained-release and immediate-release morphine with specific dosing for patients already receiving opiate therapy (Lamba, 2010 [33]); the use of different medications according to pain type (nociceptive, neuropathic or bone pain) (Long, 2020 [34], Siegel, 2017 [36]); and the use of non-opioids, weak opioids or strong opioids/weak opioids according to the WHO pain ladder (Siegel, 2017 [36], Solberg, 2015 [37]). The most used medications for pain management are short-acting opioids (31.4% of cases), followed by paracetamol (15.3%), long-acting opioids (7.1%), non-selective NSAIDs (4.8%), other (1.7%), tramadol (1.2%) and selective NSAIDs (0.9%) (Coyne, 2021 [31]).

#### 3.2.3. Included Studies’ Results

None of the included studies reported data on the effectiveness of different pain assessment strategies or on the prevalence of different types of pain; none of the included studies reported a comparison of different treatments in terms of effectiveness and safety. Bell et al. [29] reported on the percentage of medications used to treat pain and concluded that it is important to correctly identify EOL patients and initiate the appropriate targeted treatment. Chan et al. [30] demonstrated the benefit of EOL care in the ED by increasing the percentage of symptomatic treatments administered without altering the dying process. Coyne et al. [31] reported on the percentage of medications used and demonstrated the association between severe pain and the need for hospitalisation, especially in patients requiring opioids and with poor functional status. De Oliveira et al. [32] reported that palliative care needs were not met in 78.4% of patients, and only 56% of patients received medication for pain relief, showing that little attention was paid to palliative care compared to curative care. Rojas et al. [35] demonstrated the benefits of a palliative care protocol by reporting letters of appreciation from relatives.

**Table 2 jcm-12-04357-t002:** General characteristics of the included studies.

Author, Year of Publication	Type of Study	Aim of the Study	Population	Pain Assessment Tool Recommended	Drug(s) Evaluated	Results	Author’s Conclusions
Bell, 2018 [29]	Review	Review the care of geriatric patients at the end of life	Elderly patients presenting in ED	Verbal pain assessments; For non-verbal patients, facial expressions and other nonverbal cues (moaning or withdrawal) should be observed	For mild/moderate pain: Acetaminophen (1000 mg orally or rectally 3 times per day for a total of 3 g/d) For severe pain: Morphine (2 to 5 mg, IV) (or equivalent analgesic (e.g., 0.3–0.8 mg hydromorphone IV)) and dose every 15 min titrated to pain relief. In the older population, it is the recommended practice to start with the lower dose but escalate quickly (every 15 min) as needed	N/A	Although the ED may not traditionally be designed to meet palliative care, ED providers have a great opportunity to identify patients with palliative care needs and initiate a goal-oriented palliative approach that can have an impact on patient care dramatically
Chan, 2021 [30]	Retrospective cohort analysis	To study the performance of EDs in identifying patients of imminent death and the use of opioid and anticholinergic as part of symptom relief agents for patients under EOL	Adult EOL patients admitted to the ED	NR	In EOL service group, 483/688 (70.20%) received opioid and 204/688 (29.65%) received anticholinergic. In non-EOL service group, 49/95 (51.58%) received opioid and 2/95 (2.11%) received anticholinergic	The ED-based EOL care had significantly more patients receiving symptomatic treatment	Emergency physicians are capable of recognising dying patients. Emergency department-based EOL service offers adequate palliation of symptoms
Coyne, 2021 [31]	Multicentre, prospective observational study	To describe the reported pain among cancer patients presenting to the ED, how pain is managed, and how pain may be associated with clinical outcomes	Adults with active cancer presenting to the ED	Numeric Rating Scale	No drugs restriction	NSAID, nonselective in 4.8%, NSAID, selective in 0.9%, Acetaminophen in 15.3%, Tramadol in 1.2%, short-acting opioid/narcotic in 31.4%, long-acting opioid/narcotic in 7.1%, other in 1.7%	Patients who present to the ED with severe pain may have a higher risk of mortality, especially those with poor functional status. Patients who require opioid analgesics in the ED are more likely to require hospital admission and are more at risk for 30-day hospital readmission. Need of protocol-driven targeted to at-risk groups
De Oliveira, 2021 [32]	Retrospective	To identify the prevalence of PC needs in patients who die at the ED and to assess the symptom control and the aggressiveness of the care received	Adults deceased at the ED	Numeric Rating Scale	Opioids 54/117 (81.8%) patients Paracetamol 25/117 (37.9%) patients NSAISs 4/117 (6.1%) patients	384 adults died at the ED (median age 82 (IQR 72–89) years). 78.4% (95% CI 73.9% to 82.2%) presented PC needs. 3.0% (*n* = 9) were referred to the hospital PC team. 64.5% presented dyspnoea, 38.9% pain, 57.5% confusion. Dyspnoea was commonly medicated in 92%, pain in 56%, confusion in 8%	The study warns about an alarming situation of high PC needs at the ED but shows that ED clinicians likely prioritise curative-intended over palliative-intended interventions
Lamba, 2010 [33]	Review	To review the common emergency presentations in patients under hospice care	Patients in or worthy of hospice care, no age specification	NR	For patients with constant moderate to severe pain, use morphine in sustained release and immediate release forms. General principles for patients on opiate therapy: (1) calculate the morphine equivalent as a daily 24 h dose; (2) determine the breakthrough dose, which is usually 10% to 15% of this calculated daily dose; (3) titrate doses upward if pain is not controlled or more than 3 breakthrough doses are being used daily; and (4) reduce the calculated conversion dose of a new opioid by 25% to 50% when converting between different opioids because tolerance to one opioid does not imply equivalent tolerance to another because of variable opioid receptor affinity	N/A	Emergency clinicians often care for patients with a terminal illness. An understanding of hospice as a care system may increase the overall emergency clinician comfort level in discussing hospice as a care option, when appropriate, with patients and families. Using the multidisciplinary approach that is central to the hospice model may also facilitate effective management of patients under hospice care who present to the ED
Long, 2020 [34]	Review	The review provides a summary of palliative care in the ED, with a focus on the literature behind the management of EOL dyspnoea and cancer-related pain	Oncologic patients in palliative care, no age specification	Numerical rating scale, Wong–Baker FACES scale, the verbal rating scale or PAINAD	Nociceptive pain: Opioid first line for severe pain Acetaminophen or NSAIDs first line for mild to moderate pain Consider adjuvant pain medications as needed (e.g., ketamine) Neuropathic pain: Antidepressants or anticonvulsants Bone pain: NSAIDs first line for mild to moderate bone pain Severe pain will likely require opioids Add adjuvant pain medications as needed Other drugs suggested: ketamine, corticosteroids, antidepressants, anticonvulsants. In opioid-tolerant patients: administer 10 to 15% of total daily opioids dose. Reduce the dose of 25–50% in case of opioid switching or use the rapid fentanyl protocol (dose equal to 10% of total previous 24 h morphine equivalent or 50 mcg, repeat every 5 min apart and increase of 50–100% for the third administration if pain is not controlled)	N/A	The most effective therapy for cancer-related pain is opioids. For rapid alleviation of severe pain, emergency physicians may use a fentanyl rapid dose titration model. There is no literature supporting the notion that opioids hasten death at EOL
Rojas, 2016 [35]	Observational study	To develop a best practice initiative to assist actively dying patients in the ED	Actively elderly dying patients	NR	Consider a morphine IV drip, 5 mL/h, or a morphine IV bolus, 5 to 10 mg. Titrate drip every 15 min by 50% dose for comfort	401 patients evaluated by palliative care team in the emergency department. 91/401 enrolled in hospice while in the emergency department. 36/401 were enrolled in the palliative care protocol. The success of the program has been measured through letters of appreciation received from patients’ families	Care of the dying patient includes an organised interdisciplinary approach to patient care that includes open communication, medication management, and environ- mental modifications
Siegel, 2017 [36]	Review	To increase the emergency physician’s knowledge of and comfort with symptom control in palliative and hospice patients	Hospice patients, no age specification	NR	WHO analgesic ladder for nociceptive pain: mild pain (step 1) non-opioids ± adjuvants; moderate pain (step 2) weak opioids ± non-opioid ± adjuvants; severe pain (step 3) strong opioid ± non-opioid ± adjuvants Neuropathic pain: Tramadol ± Gabapentin or pregabalin ± traditional antiepileptics Bone pain: NSAIDs Evaluate morphine equivalents treating patients on chronic opioid therapy	N/A	Emergency physicians need to feel comfortable with the use of morphine and morphine equivalents in managing pain both in the ED and when discharging patients home; palliative care benefits both the patient and the medical system at large: patients report a better quality of life and higher satisfaction when receiving palliative care services and medical costs are decreased.
Solberg, 2015 [37]	Review	To review the approach to palliative care in the Emergency Department	Patients worthy of palliative care, no age specification	NR	WHO analgesic ladder for nociceptive pain: mild pain (step 1) non-opioids ± adjuvants; moderate pain (step 2) weak opioids ± non-opioid ± adjuvants; severe pain (step 3) strong opioid ± non-opioid ± adjuvants	N/A	Currently, there is little research and evidence on the use of PC in the ED. However, one may infer patient satisfaction: other outcomes will improve and costs will be reduced, as they have in the inpatient setting. The systematic integration of PC into the ED is slowly occurring, as it is a recognised subspecialty of emergency medicine

Note: ED: emergency department; IV: intravenous; NA: not admitted; NSAIDS: non-selective anti-inflammatory drugs; NR: not reported; PC: palliative care; WHO: World Health Organisation.

## 4. Discussion

Dealing with terminally ill patients in the emergency department has become increasingly common. As mentioned earlier, such patients have complex needs, and it can be challenging to make a decision that is acceptable to both the patient and the caregivers. However, the 2005 “American College of Emergency Physicians policy statement on end-of-life care in the emergency department” states that emergency physicians should respect the dying patient’s need for care, comfort, and compassion; communicate promptly and appropriately with patients and their families about end-of-life care decisions; inquire about the patient’s goals of care; and respect the dying patient’s wishes, including those expressed in living wills, whenever possible [38]. According to the Montreal Declaration, access to pain treatment is a fundamental human right for all people, including terminally ill patients, and denial of pain treatment is “profoundly wrong and leads to unnecessary and harmful suffering” [39]. Only nine of the 532 studies identified by the search criteria were included in the present study, and none of them addressed the efficacy and safety of different approaches to pain management. As shown in Figure 1, despite the rigorous search strategy, most studies were excluded because they addressed topics other than end-of-life pain management, and only a few other studies included EOL patients. Most of these studies illustrated the causes of ED visits for EOL patients, the need for palliative care, factors influencing the possibility of palliative care pathways, and their global impact in the ED, but they lacked data on pain management in EOL patients. The included observational studies reported on the percentage of patients admitted with pain, the percentage of patients receiving pain treatment, and the type of medication used (Chan, 2021 [30], Coyne, 2021 [31], de Oliveira, 2021 [32], Rojas, 2016 [35]). Rojas et al. [35] reported the effectiveness of a particular protocol based on an increase in the number of letters received from families; other measures of effectiveness were not reported. The included reviews (Bell, 2018 [29], Lamba, 2010 [33], Long, 2020 [34], Solberg, 2015 [37], Siegel, 2017 [36]) were heterogeneous in terms of proposed medications and overall approach. Bell et al., 2018 [29], discussed the need for ED clinicians to recognise the need for palliative care, communicate effectively and manage symptoms in actively dying patients. Pain should be assessed verbally or “via facial expressions and other non-verbal cues”, and it is recommended that paracetamol or morphine and equivalents be administered without referring to non-pharmacological pain management or other medications. Lamba et al. [33] discussed the optimal approach to identifying patients eligible for palliative care and the appropriate management of these patients. Pain management is discussed, noting that there is no upper limit or maximum recommended opioid dose, and recommending the use of a fixed dosing regimen combining sustained-release medications with immediate-release medications and the use of web resources to manage opioid doses in opioid-tolerant patients suffering from both acute and chronic uncontrolled pain. There were no recommendations regarding the appropriate pain scale or the use of non-pharmacological or other medications. Long et al. [34] provided an interesting overview of palliative care in ED with a focus on cancer-related pain in 2020. They also discussed the importance of living wills in palliative care, including do not resuscitate and do not intubate orders, effective communication with patients using appropriate discussion guides, and pain management. Several pain assessment tools have been proposed, including the Numeric Rating Scale (NRS), the Wong–Baker Facial Scale (VRS) for cooperative patients and the PAINAD (Critical Care Pain Observational Tool) for non-communicative patients. Long et al. also address the different types of pain and recommend the use of opioids (fentanyl, morphine, and hydromorphone) in appropriate doses for opioid-tolerant patients and non-opioid medications (including NSAIDs, paracetamol, ketamine, and corticosteroids), especially for patients with neuropathic pain, while also discussing the use of antidepressants and anticonvulsants [34]. Siegel et al. reported on “the ABC of symptom management for emergency physicians in palliative care” [36]. They named the 12 generalist palliative care skills expected of all emergency physicians, according to Quest et al.: assessing the severity of illness/dying, formulating a prognosis, difficult communication (breaking bad news, disclosure of death, living wills and planning), family presence during resuscitation, management of pain or non-pain symptoms, withdrawal or refusal of unhelpful care, dealing with the dying, dealing with hospice patients or referral to palliative systems, ethical and legal issues, spiritual/cultural competence and dealing with the dying child [40]. 

### 4.1. Management of Pain at EOL in the ED in Clinical Practice

#### 4.1.1. Identify Patient Worthy of End-of Life Care

Palliative or end-of-life care may be necessary for ED patients in two circumstances: (i) following a likely fatal event (major stroke, cardiac event, or surgical emergency) and (ii) in advanced chronic illness with progressive symptom burden and an unmet need for palliative care. In both cases, patients and their families may not be aware that they are dying. Several screening tools have been developed to identify patients in the emergency department who are eligible for palliative care consultation [41], which is based on (i) functional status, such as the Karnofsky Performance Status Scale [42,43,44], the Knau classification [45] and the Katz Activities of Daily Living [46,47], the Timed Up and Go test (risk of falling) [46], the Functional Assessment Staging Tool [43], the Outcome and Assessment Information Set [48] or the Palliative Performance Scale [48]), (ii) comorbidities, such as the Charlson Comorbidities Index [45,47], Charlson–Manitoba Comorbidity [49], (iii) short-term mortality, such as the Modified CriSTAL [50,51], identification of at-risk seniors [46,47,49], and the “Surprise Question” [52], or (iv) a combination of them, such as the NECessidades PALiativas Centro Colaborador da Organização Mundial da Saúde-Institut Català d’Oncologia (NECPAL CCOMS-ICO tool) [53] or George et al.’s simple checklist to evaluate the terminal illness and severe uncontrolled symptoms [54]. However, no screening tool has been shown to be better than another [55]. Despite several screening tools, a clear conversation with patients may be the better approach to reveal the presence of hidden palliative care needs. Reuter et al. in 2019 proposed the 5-SPEED survey to identify patients in the ED with unmet palliative care needs [56]: (i) How much discomfort do you feel? (ii) How challenging is it for you to receive the care you require at home? (iii) What challenges do you encounter when taking medication? (iv) How much do you feel overwhelmed? (v) How challenging do you find it to receive medical care that meets your needs? Moreover, to tailor the extent of medical intervention to a patient’s goals, it is essential to discuss the patient’s current condition and treatment objectives [57]. In his 2015 book “Being Mortal”, Atul Gawande proposed five questions that can serve as a framework for such clear conversations: “What do you know about your illness and condition? What are your future-related fears and concerns? What are your priorities and objectives? What outcomes did you deem acceptable? What are you willing and unwilling to sacrifice? What would a good day look like in the future?” [58].

Due to poorly controlled symptoms or conflicts between the patient (who may refuse life-prolonging measures) and carers (who insist on life-prolonging measures), terminal patients are occasionally referred to the emergency department. In addition, referral to ED may indicate that the patient or family is anxious and unable to cope with the distressing symptoms of impending death. According to Reeves [59], when a patient is mistakenly admitted to the emergency department (ED), it is imperative to determine resuscitation status and treat distressing signs by providing supportive care and avoiding unnecessary tests and invasive procedures, providing privacy, and discussing hospitalisation with the patient, family, and palliative care service.

#### 4.1.2. Pain Assessment in EOL Patients in ED

Pain assessment is critical to pain management and should include information about the location, quality, intensity, onset, duration, and frequency of pain as well as factors that relieve or exacerbate pain. Cancer-related pain should be classified according to the ICD-11 taxonomy and the intensity and impact of pain on quality of life determined [60]. The distinction between peripherally generated pain and centrally maintained pain may be more accurate in assessing pain and selecting the appropriate drug treatment for the patient [61]. Pain rating scales adapted to the patient’s age and cognitive status help standardise treatment and provide objective assessment tools. Acute pain can be measured with the Numerical Rating Scale (NRS), the Visual Analogue Scale (VAS) [62] and the Faces Pain Scale Revised [63].

However, chronic pain often affects patients at the end of life, so a multidimensional assessment may be more appropriate. The Brief Pain Inventory (BPI) [64] measures actual and the past 24 h pain intensity, multiple sources of pain and the impact of pain on daily activities. The Short-Form McGill Pain Questionnaire (SF-MPQ) [65] measures sensory, affective, and general pain in patients with chronic pain. The presence of neuropathic pain should be thoroughly assessed using appropriate scales such as the Leeds Assessment of Neuropathic Symptoms and Signs (LANSS) [66] or the Neuropathic Pain Scale (NPS) [67], which assesses the presence of stabbing, burning, numb, electric, tingling, pressing, freezing and simple touch pain. Scales that assess facial expression, body movements, crying and well-being, such as the Face Leg Activity Cry Consolability (FLACC) for infants and children [68], the Pain Assessment in Advanced Dementia (PAINAD) for non-verbal patients [69] and the Critical Care Pain Observation Tool for non-verbal patients [70], should be used to identify and manage pain in these patients. A summary of the available pain scales can be found in Table 3.

#### 4.1.3. Non-Pharmacological Management of Pain in EOL Patients

The non-pharmaceutical treatment of end-stage pain remains under-researched and under-utilised. Consistent with the biopsychosocial model of pain, non-pharmacological techniques are essential to a comprehensive pain management strategy, often require minimal resources, and they can be implemented in the emergency department, although it is busy and often chaotic [71,72]. Managing the psychological and social aspects of pain (including an appraisal of thoughts and emotions, perceptions of pain, beliefs and attitudes, unhelpful thinking patterns, previous pain experiences, fear, depression, anxiety, psychological distress, and sharing information at caring level) can have a significant impact on patient suffering [73]. Sharing information about stressful medical procedures, pain and postoperative pain has been shown to be beneficial [74,75]. Despite limited evidence of benefit in the emergency department [71], the “difficult conversation” (i.e., the communication of bad news and prognosis) is an essential part of patient care. Detailed conversations can still be held with people with dementia to communicate their wishes, preferences, and choices, even though dying people are often cognitively or emotionally unable to have such a conversation, which should ideally take place before death. However, a culture change is needed in the health professions so that such conversations may be seen as “essential” rather than “difficult”. This conversation is a professional duty and a right for those individuals and families who want it [76]. Baile et al. [77] created the SPIKES model in 2020, which is a useful mnemonic model consisting of six components for communicating bad medical news: −Setting: creating time, space, and resources for appropriate, uninterrupted communication with patients and their families.−Perception: asking first about illness awareness and severity.−Invitation: ensuring that the patient or family members are ready to discuss palliative or end-of-life care.−Knowledge: ensure that the patient or family members are aware of the disease context in which the pathology manifests.−Emotion: address emotions empathically; use the mnemonic NURSE [78] (Name the patient’s emotion, Understand the emotion through empathy, Respect the patient’s response, Support the patient, Explore the patient’s response, and inquire about the emotion).−Strategy: develop a plan for shared care. 

Non-pharmacological interventions (such as massage, aromatherapy, and music therapy) to promote well-being or other aspects related to well-being such as stress, fatigue, anxiety, and depression can be beneficial in palliative care [79] and even in the management of breakthrough pain in cancer [80]. Other techniques such as transcutaneous electrical nerve stimulation (TENS), acupuncture, cold and heat, traction and patient positioning all play a role in acute pain, especially traumatic pain; however, there is no evidence to support their use in the emergency setting in EOL patients [73]. The environment is critical to non-pharmaceutical treatment and care. When patients arrive at the emergency department and death is imminent, they and their families should be moved to a private, quiet room. The number of monitors should be reduced as much as possible, and the presence of family members and caregivers should be allowed. Some emergency departments have established single-bed rooms for end-of-life care to ensure respect and dignity for the patient and to allow family members to stay with their loved ones around the clock in a quiet environment [30,81].

#### 4.1.4. Pharmacological Management of Pain in EOL Patients

The three-step WHO pain ladder is a benchmark for the selection of pharmacological agents for pain treatment [82]. This strategy was proposed in 1986 to provide adequate analgesia to cancer patients. However, in the years that followed, this technique was used to treat pain of any origin. In accordance with the WHO pain ladder, patients have prescribed medication according to the severity of their pain: (i) non-steroidal anti-inflammatory drugs (NSAIDs) or paracetamol with or without additives for level I, mild pain; (ii) weak opioids such as hydrocodone, codeine and tramadol with or without non-opioid analgesics and with or without adjuvants for level II, moderate pain; (iii) strong opioids such as morphine, methadone, fentanyl, oxycodone, buprenorphine, tapentadol, hydromorphone or oxymorphone, with or without non-opioid analgesics and with or without adjuvants, for level III, severe pain. Antidepressants (such as tricyclic antidepressants (TCAs) or serotonin-norepinephrine reuptake inhibitors (SNRIs)), anticonvulsants (such as gabapentin and pregabalin), topical anaesthetics (such as lidocaine patches), topical therapies (such as capsaicin), and corticosteroids are examples of adjuvants. The WHO pain ladder is considered one of the most successful global health interventions [83,84] and one of the most significant advances in the treatment of pain patients, as it is effective in 69 to 100% of patients [85,86]. However, numerous authors have criticised this approach for the following reasons: (i) Some adjuvants are first-line therapies for certain types of pain, such as neuropathic pain [87]; (ii) Weak opioids have been shown to be of little use in the treatment of pain [88]; (iii) The presence of NSAIDs proposed for tier I gives the false impression that these drugs are safe in the treatment of pain, while they have been shown to be associated with numerous adverse effects [89]. In addition, NSAIDs may have a role in the treatment of bone pain due to their effect on prostaglandin production. However, recent systematic reviews have found little evidence to support or refute the use of NSAIDs in this area [90,91]; (iv) Specific types of pain such as neuropathic pain, bone pain or fibromyalgia, for which opioids are of limited benefit, have not been considered; (v) The benefits of interventional and minimally invasive procedures [92], as well as the use of complementary and alternative medicine, relaxation techniques, psychological support and physiotherapy [93], have not been considered. Morphine is the first-line treatment for severe pain, but it is poorly available orally, the dose required varies widely, active metabolites can accumulate and lead to neurotoxicity, and morphine-induced histamine release can lead to poorly tolerated side effects, whereas oxycodone, hydromorphone, and fentanyl have the same efficacy with fewer side effects and better tolerability [94]. Neuropathic pain must be appropriately identified and treated, and opioids should only be considered as a third-line treatment due to their poor efficacy. Although tramadol is a second-line agent and gabapentin, duloxetine and antidepressants are recommended as first-line agents for neuropathic pain; these drugs take several days to work and there is limited evidence of their efficacy and safety, especially in the ED [95]. In addition, antiepileptic drugs such as phenytoin, lamotrigine, valproic acid and levetiracetam have not shown efficacy despite increased side effects [96,97,98]. Ketamine inhibits glutamatergic neurons primarily through its antagonistic action on NMDA receptors. However, its action on dopaminergic, adrenergic, serotoninergic, opioid, and cholinergic receptors, as well as on spinal GABA interneurons, appears to be responsible for the analgesic effect of this drug, which is effective in chronic, neuropathic and cancer pain [99]. In addition, corticosteroids, and magnesium sulphate [100,101,102] have been shown to be useful adjuvants, especially in neuropathic pain and bone pain [103,104,105]. Therefore, the most effective drug for neuropathic pain needs to be identified [94], especially for acute pain management in the emergency department, so a multidisciplinary approach is strongly recommended.

See Table 4 for a summary of suggested drugs for pain management, indicating the type of pain, drug, mechanism of action, dosage, onset of action, peak effect, and duration [106,107].

Interventional pain management may be an option for patients with severe, persistent pain who do not respond to other pharmacological treatments. However, it should be integrated into a multidisciplinary approach to pain management and not just considered as a last resort [108]. Interventional pain management includes injection therapies, including soft tissue and joint injections, as well as spinal-related injections (i.e., epidural steroid injections, facet joint injections, facet denervation approaches and sacroiliac injections), non-neurolytic (including sympathetic blockade of the stellate ganglion, lumbar sympathetic truncus, celiac plexus, superior hypogastric plexus and impar ganglion or somatic nerve blocks, including paravertebral and intercostal nerve blocks, brachial plexus block and epidural or intrathecal blocks), neurolytic blocks (neurolysis of the celiac plexus, neurolysis of the superior hypogastric plexus and dorsal punctate midline myelotomy, especially for visceral intractable pain) or newer advanced neuraxial techniques, including spinal cord stimulation. Interventional pain procedures provide effective pain relief, reduce symptom burden, decrease opioid use, and have a favourable safety profile [109,110,111]. Although some peripheral and fascial nerve blocks have been shown to be safe and feasible in the emergency department [112,113,114], they are still underused, and other invasive treatments should be discussed on a case-by-case basis in a multidisciplinary team. 

#### 4.1.5. Special Situations: Rapid Pain Worsening in Chronic Pain

Opioids are increasingly used to treat chronic pain in terminally ill patients. Patients currently receiving opioids should be assessed for the amount of opioids they were taking daily before the onset of the new pain, and appropriate doses of opioids should be prescribed to treat the baseline pain in combination with short-acting opioids to treat the new severe pain. Equianalgesia refers to the amount of different opioid formulations and/or different routes of administration that achieve the same analgesic effect (the morphine milligram equivalents, MME), while opioid titration refers to the individual adjustment of the drug dose [115]. There are numerous equianalgesic dosing charts or web/application resources available to help clinicians manage severe pain in opioid-tolerant patients or switch formulations and routes of administration. A dose equivalent to 1/6 of the daily dose of the first opioid should be administered when a second opioid is added to ongoing treatment [115].

##### Breakthrough Pain

Davies et al. [80] define breakthrough pain as a transient exacerbation of pain that occurs spontaneously or in response to a specific predictable or unpredictable trigger, although the background pain is relatively stable and well-controlled. It is common in patients with cancer pain (40–80%), but there is currently no evidence of its prevalence in chronic non-cancer pain [116]. Transmucosal fentanyl preparations are reserved for situations requiring a faster onset and shorter duration of action [117]. In patients who are not opioid-naïve, the right opioid dose should be calculated as follows [115]: (i) determine the last 24 h effective MME; (ii) determine the breakthrough dose, which is usually 10–15% of the calculated MME daily dose; (iii) titrate the dose upwards if the pain is not under control or if more than 3 breakthrough doses per day are administered; and (iv) start with one opioid (via continuous infusion, oral or transdermal patch) and reduce the calculated conversion dose of a new opioid by 25% to 50% when switching from one opioid to another, as tolerance to one opioid is not the same as tolerance to another. Long et al. [34] suggested a rapid fentanyl titration protocol in which intravenous fentanyl boluses are administered 5 min apart until the pain is controlled, with the initial dose equal to 1/10 of the total MME of the previous 24 h if the reported pain is at least 4/10 on the NRS (or equivalent according to the pain scale used), and the fentanyl dose is increased by 50–100% from the third administration.

##### Opioid-Induced Hyperalgesia

Opioid-induced hyperalgesia is characterised by increased pain sensitivity despite a higher opioid dose, which is often accompanied by a diffuse extension of the pain site and allodynia. It can occur at any dose but is more pronounced at higher parenteral doses of morphine and hydromorphone. The initial treatment consists of reducing or eliminating the current opioid dose or switching to an opioid with less neurotoxic effects, such as fentanyl, and maximising non-opioid adjuvants when indicated. The use of low-dose ketamine for pain relief may have a role in the management of this issue in the ED [99].

#### 4.1.6. Palliative Sedation

Although pain symptoms in EOL patients often respond to pharmacological and/or non-pharmacological interventions, there may be cases where treatment is extremely difficult, impossible, or ineffective. This is the case with so-called refractory symptoms because (i) the treatment does not respond, (ii) the analgesic effect is too long in coming, or (iii) the patient cannot tolerate the side effects [118,119,120,121]. Palliative sedation is defined as an intervention aimed at relieving intolerable suffering caused by one or more refractory symptoms [122] and is limited to terminally ill patients. Sedation in palliative care is fundamentally different from euthanasia. Unlike euthanasia, which aims to bring about the patient’s death through standard protocols, palliative sedation aims to relieve intolerable suffering through appropriate sedation. Furthermore, palliative sedation does not hasten death, which may be an important factor for physicians and families to consider when prescribing this treatment [123,124,125], so the term “terminal sedation” is no longer appropriate [119]. Sedation in palliative care can be light, deep, continuous, or intermittent [126]. Short-term sedation and intermittent sedation can be used at an earlier stage to provide temporary relief while another therapeutic approach takes effect. When light sedation is ineffective, when the condition is severe and unresponsive to treatment, when death is expected within hours or days and the patient specifically requests it, or in the case of a catastrophic end-of-life event, deeper sedation should be used [122]. Palliative sedation is based on the use of sedatives selected on a case-by-case basis and titrated to the minimally effective dose by continuous reassessment using validated scales such as the Richmond Agitation–Sedation Scale [118,126,127,128]. Because of its short half-life, midazolam is recognised in all guidelines as the drug of choice for palliative sedation. Alternatives to midazolam include diazepam, lorazepam, and clonazepam [118,122]. Sedation can be achieved with midazolam (1–5 mg bolus, 1–5 mg/h via continuous infusion), lorazepam (1–5 mg bolus, 0.025–0.05 mg/kg/h via continuous infusion) or propofol (20 mg to 1 mg/kg bolus, 5–10 mg/h to 0.5–3 mg/kg/h via continuous infusion) [122,128,129,130]. However, there is no approved maximum dose, and the dosage must be adjusted according to the degree of sedation and pain control. In addition, isolated case reports have been published on the successful use of dexmedetomidine in palliative care for cancer pain and opioid-induced hyperalgesia [131]. The administration of a neuroleptic (preferably levomepromazine, but also chlorpromazine, clotiapine and promethazine) may be considered [118,122], especially in the presence of concurrent agitation due to suspected delirium. Opioids should be administered as pain and dyspnea medications but not as sedatives. Morphine is the opioid of choice, but other opioids have been reported in the literature [132]. The dissociative anaesthetic ketamine maintains a clear airway and stabilises the cardiovascular system. However, further research is needed to determine which terminally ill patients may benefit from ketamine therapy [133].

### 4.2. Limitations of the Present Narrative Review

For the purposes of this review, all studies were considered that addressed pain management in terminally ill, EOL, palliative or hospice patients admitted to the emergency department. However, some limitations due to the inclusion criteria need to be discussed. First, despite the broad inclusion criteria, only a few observational studies were included, and more rigorous, high-quality studies are needed to investigate the efficacy and safety of actual pain management in this specific category of patients. Second, the included studies were based on data obtained in settings other than the emergency department, so data on effectiveness in this specific setting may vary. Third, the included studies were on adult patients, while no data were available for the paediatric population. Fourth, because of the heterogeneity and small number of studies, it was not possible to conduct a meta-analysis and make a definitive statement about the best method to ensure effective pain management. Fifth, studies, guidelines and recommendations from societies that were not in English were excluded, which could potentially alter the conclusions of this review.

## 5. Conclusions

Pain management is a fundamental component of patient care at every stage of life, regardless of the causes of pain, concomitant diseases, state of consciousness, and life expectancy. Although pain is a leading cause of admission to the emergency department, oligoanalgesia is a common, deeply inexcusable phenomenon that should be vigorously addressed through research, the education of healthcare professionals, and the establishment of appropriate treatment pathways. Emergency departments are often crowded, chaotic environments with a mismatch between medical staff and patients admitted with unique organisational and therapeutic resources. Despite intensive research in palliative care on the management of end-of-life symptoms, there is limited data on how to better manage palliative care patients and terminally ill patients admitted to the emergency department with severe pain. The identification of patients eligible for palliative and hospice care, the identification of unmet needs for palliative care, patient-centred multidimensional pain assessment, and a multidisciplinary approach are critical steps in the care of patients in the ED. Further research is needed in these areas to provide the best evidence-based supportive therapy. Although strong opioids remain the cornerstone of treatment for severe pain, ‘total pain’ has multiple components, and appropriate treatment should be carefully selected for the specific type of pain, including non-pharmacological and pharmacological pain management and sedative palliation.

## Figures and Tables

**Figure 1 jcm-12-04357-f001:**
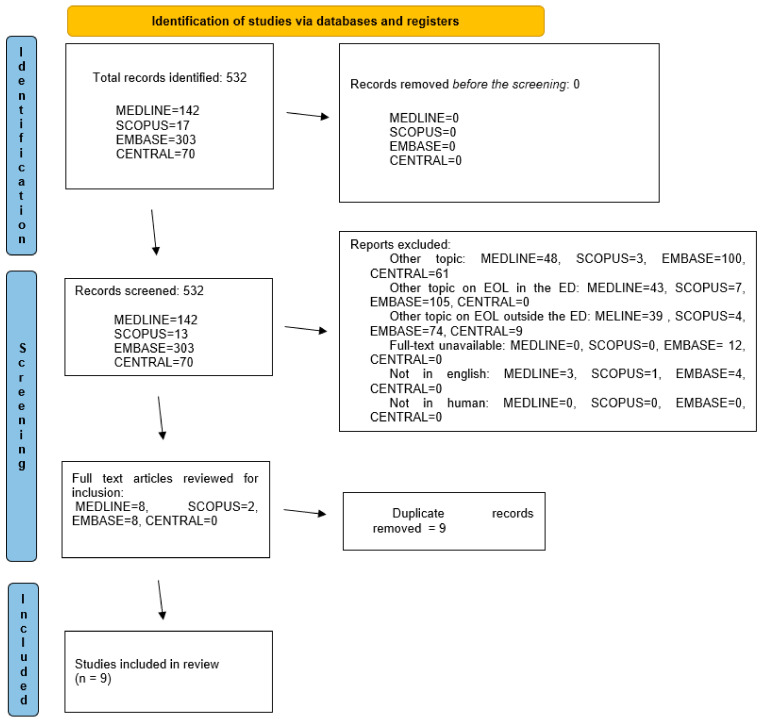
PRISMA 2020 flow diagram for identification, screening, and study inclusion.

**Table 1 jcm-12-04357-t001:** Pain terms and definitions according to the International Association for the Study of Pain.

Terms	Definition
Pain	An unpleasant sensory and emotional experience associated with, or resembling that associated with, actual or potential tissue damage.
Allodynia	Pain to stimulus that does not normally provoke pain, leading to an unexpectedly painful response.
Hyperalgesia	Increased pain from a stimulus that normally provokes pain.
Neuralgia	Pain in the distribution of a nerve or nerves.
Neuropathy	A disturbance of function or pathological change in a nerve or in several nerves.
Nociception	The neural process of encoding noxious stimuli.
Nociceptive pain	Pain that arises from actual or threatened damage to non-neural tissue and is due to the activation of nociceptors.
Neuropathic pain	Pain caused by a lesion or disease of the somatosensory nervous system.
Central neuropathy pain	Pain caused by a lesion or disease of the central somatosensory nervous system.
Peripheral neuropathic pain	Pain caused by a lesion or disease of the peripheral somatosensory nervous system.
Nociplastic pain	Pain that arises from altered nociception despite no clear evidence of actual or threatened tissue damage causing the activation of peripheral nociceptors or evidence for disease or lesion of the somatosensory system causing the pain.

**Table 3 jcm-12-04357-t003:** Scales for pain recognition in patients at EOL in the ED.

	Scale	Items Evaluated
Unidimensional pain evaluation:		
	Numerical Rating Scale, NRS	0 for “no pain” to 10 for “worst pain”, verbally reported
	Visual Analogue Scale, VAS	A scale from 0 to 10 is reported on a paper sheet, and the patient needs to mark the level of pain in this scale
	Faces Pain Scale Revised	6 different facial expressions (from a smile to intense crying) are depicted and the patients should choose the most appropriate to describe pain intensity
Multidimensional pain evaluation:		
	The Brief Pain Inventory, BPI	1. Presence of other types of pain 2. Sign areas of pain in the body With a number from 0 to 10: Rate the worst pain in the last 24 h Rate the least pain in the last 24 h Rate the average pain Rate the actual pain Enlist the medications used for pain management Rate how much relief pain medication provided Rate how much pain interfered with: general activity, mood, walking, normal work, relations with other people, sleep, enjoyment of life.
	The Short-Form McGill Pain Questionnaire, SF-MPQ	Compiled by the patients, which has to rate as none (0), mild (1), moderate (2) more severe (3) the presence of: Throbbing, Shooting, Stabbing, Sharp, Cramping, Gnawing, Hot/burning, Aching, Heavy, Tender, Spitting, Tiring/Exhausting, Sickening, Fearful, Punishing/Cruel
	Edmonton Symptom Assessment System	Compiled by the patient, who has to rate from 0 to 10 how he felt in the last 24 h about: pain; tired, nauseated, depressed, anxious, drowsy, appetite, well-being, shortness of breath, abdominal discomfort, able to move normally
Neuropathic pain:		
	Leeds Assessment of Neuropathic Symptoms and Signs (LANSS)	Evaluate the presence or absence of: (I) unpleasant sensations described as pricking, tingling or pins in skin in pain area, skin abnormally sensitive to touch, pain coming suddenly and in burst, sensation of skin temperature altered; (II) objectively tasted: presence of allodynia (different reactions to soft touch in painful and non-painful areas); altered pinprick threshold
	Neuropathic Pain Scale (NPS)	The patient has to rate via VAS the presence of: burning pain; overly sensitive to touch, shooting pain, numbness, electric pain, tingling pain, squeezing pain, freezing pain, how unpleasant is usual pain, how overwhelming is usual pain
Non-cooperative or cognitively impaired patients		
	Face Leg Activity Cry Consolability, FLACC	Observer scores from 0 to 2 for the presence of abnormal: face expression, legs position, activity, cry, consolability
	Pain Assessment in Advanced Dementia, PAINAD	Observer scores from 0 to 2 for the presence of abnormal: breathing, negative vocalisation, facial expression, body language, consolability
	Critical Care Pain Observation Tool	Observer scores from 0 to 2 for the compliance with the ventilator in case of an intubated patient, facial expression, body movements, muscle tension

Note: The different scales are presented only for informative purposes. Please refer to the original article for the correct application of the scale in clinical practice.

**Table 4 jcm-12-04357-t004:** Summary of available drugs for pain management in EOL patients in the ED.

	Medication	Mechanism of Action	Dosing	Onset, Peak of Effect, Duration
Nociceptive pain				
Mild				
	Acetaminophen	Unclear mechanism of action, activation of descending serotonergic inhibitory pathways in the CNS may be a component, COX-1, COX-2, CO-3 inhibition	IV: maximum dose of 4 g/day, 1 g every six hours Patients <50 kg or with chronic alcoholism, malnutrition, or dehydration: 12.5 mg/kg every four hours or 15 mg/kg every six hours, maximum 750 mg/dose, maximum 3.75 g/day	Onset: oral <1 h, IV: 5–10 min Peak of action: UV; 1 h Duration: oral, IV: 4–6 h
Nonsteroidal anti-inflammatory drugs	Ibuprofen	COX-1 and COX-2 reversible inhibition; reduced prostaglandin formation	PO 200 to 800 mg 3–4 times daily IV 400 to 800 mg every 6 h as needed, up to 3200 mg in acute phase	Onset: PO: 30–60 min Peak: PO: ND Duration:: PO: ND
	Ketorolac		PO ≥ 50 kg: 20 mg PO, followed by 10 mg every 4–6 h as needed IV ≥ 50 kg: 30 mg IV as a single dose or 30 mg every 6 h	Onset: 30 min Peak: 2–3 h Duration: 4–6 h
COX-2 selective NSAIDs	Celecoxib	COX-2 inhibition, with decreased prostaglandin precursors formation	PO: 200 mg daily or 100 mg every 12 h, maximum dose: 400 mg per day	Onset: ND Peak: capsule: 3 h, oral solution: 0.7–1 h Duration: ND. Half-life elimination: 0.7–1 h
	Eterocoxib		PO: 30 to 60 mg once daily, maximum dose: 120 mg in acute pain, up to 8 days, 60 mg otherwise	Onset: ND Peak: 1 h Duration: ND
Moderate				
Opioids	Codeine	Mu-, delta-, kappa-opioid receptors agonist; inhibition of ascending pain pathways and altered perception and response to pain	30 to 60 mg every 4 to 6 h as needed	Onset: 0.5–1 h Peak: 1–1.5 h Duration: 4–6 h
	Tapentadol	Mu-opioid receptors agonist, sodium-dependent noradrenaline transporter inhibitor; inhibition of ascending pain pathways and altered perception and response to pain	PO 50 to 100 mg every 4 to 6 h as needed; maximum total daily dose: 600 mg/day.	Onset: 1.25 h Peak: 1.25 h, long acting formulations: 3–6 h Duration: ND, half-life elimination: 4 h, long acting formulations: 5–6 h Peak: Immediate release: 1.25 h; Long-acting formulations: 3–6 h Duration 4–6 h
Severe				
Opioids	Fentanyl	Mu-, delta-opiate receptors agonist at many sites within the CNS; increases pain threshold, alters pain perception and inhibits ascending pain pathways	IN: 1.5–2 mcg/kg IM: 50–100 mcg every 1–2 h as needed (only if IV not available) IV: 50–100 mcg every 30–60 min (1.0 mcg/kg) or 1–3 mcg/kg Acute pain in patients on chronic opioid therapy (e.g., breakthrough cancer pain): transmucosal: buccal tablet: 100 mcg, IN: 100 mcg, sublingual spray: 100 mcg, IV: see main text for specific dosing	Onset: IN: 5 to 10 min; IM: 7 to 8 min; IV: Almost immediate; Transdermal patch (initial placement): 6 h; Transmucosal: 5 to 15 min. Peak: Intranasal: 15–21 min Transdermal patch: 20–72 h Duration: IM: 1 to 2 h; IV: 0.5 to 1 h; Transdermal (removal of patch/no replacement): Related to blood level; some effects may last 72 to 96 h due to extended half-life and absorption from the skin, fentanyl concentrations decrease by ~50% in 20 to 27 h; Transmucosal: Related to blood level
	Morphine	Mu-, kappa-, delta-opiate receptors agonist; inhibition of ascending pain pathways, alters perception and response to pain, CNS depression	IV: 4 mg every 2 h (0.1 mg/kg), 2–5 mg IV every 15 min titrated to pain relief Acute pain in patients on chronic opioid therapy (e.g., breakthrough cancer pain): calculate the 24 h MME using appropriate opioids equivalent chart, administer 15% of the total dose IV. Repeat as needed	Onset: Oral (immediate release): ~30 min; IV: 5 to 10 min. Peak: Oral: 1 h, IM: 30–60 min, IV: 20 min, SUBQ: 50–90 min Duration: Immediate-release formulations (tablet, oral solution, injection): 3 to 5 h, extended release: 8–24 h; IV half-life elimination: 2–4 h
	Hydromorphone	Mu-, kappa-opioid receptor agonist, delta-opioid receptor partial agonist	IV: 0.4–1 mg every 2–4 h (mg/kg)	Onset: Oral: 15–30 min, IV: 5 min Peak: Oral: 30–60 min, IV: 10–20 min Duration: 3–4 h
	Oxycodone	Mu-, kappa-, delta-opioid receptor	Oral: Initial: 5 mg every 4 to 6 h as needed Usual dosage range: 5 to 15 mg every 4 to 6 h as needed Acute pain in patients on chronic opioid therapy (e.g., breakthrough cancer pain): Immediate release: Oral. Usual dose: In conjunction with the scheduled opioid, administer 5% to 15% (rarely up to 20%) of the 24 h oxycodone requirement (or morphine milligram equivalents) as needed using an immediate release formulation every 4 to 6 h with subsequent dosage adjustments based upon response	Onset: 10–15 min Peak: 0.5–1 h Duration: 3–6 h
Non-opioids	Ketamine	Glutamate receptor ionotropic NMDA 3A, antagonist, 5-hydroxytryptamine receptor 3A potentiator, alpha-7 nicotinic cholinergic receptor subunit antagonist, cholinesterase inhibitor, nitric oxide synthase inhibitor; analgesia, modulate central sensitisation, hyperalgesia and opioid tolerance	IV: 0.1–0.3 mg/kg bolus; 0.1–0.3 mg/kg/hours via continuous infusion	Onset: IV: 10–15 min, IN: 10 min Peak: IN: 10–15 min, Duration: IV: 30 min, IN: 60 min.
Neuropathic pain				
First-line therapy				
Antiseizure drugs	Gabapentin	Voltage-dependent calcium channels subunit alpha-2-delta-1 and delta-2 presynaptically located throughout the brain inhibitor, modulating the release of excitatory neurotransmitters which participate in epileptogenesis and nociception	PO: 100–1200 mg	Onset: ND Peak: 2–4 h Duration: ND
	Pregabalin	Voltage-dependent calcium channel subunit alpha-2/delta-1, effect not known, inhibiting excitatory neurotransmitter release including glutamate, norepinephrine, serotonin, dopamine, substance P, and calcitonin gene-related peptide	PO: 75–300 mg BID	Onset: pain management efficacy may be noted as early as the first week of therapy. Peak: 1.5–3 h Duration: ND
Antidepressants				
SNRI	Duloxetine	Serotonin and norepinephrine reuptake and dopamine reuptake inhibitor	PO: 30 mg daily for 7 days, then 60 mg daily for chemotherapy-induced peripheral neuropathy, 60 mg daily up to 120 for diabetes mellitus induced neuropathic pain	Onset: ND Peak: ND Duration: ND
	Venlafaxine	Serotonin and norepinephrine reuptake inhibitor and weak dopamine reuptake inhibitor	PO: 15–100 mg	Onset: ND. Onset of action: pain management effects may be noted as early as the first week of therapy Peak: ND Duration: ND
Tricyclic antidepressants	Amitriptyline	Inhibit norepinephrine and serotonin reuptake, block sodium and calcium channels and NMDA receptors	PO: Initial: 10 to 25 mg once daily at bedtime; may gradually increase dose based on response and tolerability in 10 to 25 mg increments at intervals ≥ 1 week up to 150 mg/day given once daily at bedtime or in 2 divided doses	Onset: ND Peak: 2–5 h Duration: ND
Second-line therapy				
	Capsaicin 8% patch	Transient receptor potential vanilloid 1 receptor (TRPV1) agonist via a nociceptor defunctionalisation via a reduction in TRPV1 expression in nerve endings due to capsaicin stimulation	0.025–0.1% transdermal patch	Onset: 2–4 weeks in continuous therapy Peak: ND Duration: ND
	Lidocaine patch	Nerve conduction blockage via a reduction in permeability to sodium ions	-	Onset: 4 h Peak: ND Duration: ND
	Tramadol	Mu-opiate receptor agonist, serotonin and norepinephrine reuptake inhibitor with inhibition of ascending pain pathways and altered perception and response to pain	PO 50–100 mg every 4–6 h, IV 50–100 mg every 4–6 h max dose 400 mg daily	Onset: < 1 h Peak: 2–3 h Duration: 6 h
Third-line therapy				
See strong opioids				
Other drugs suggested as adjuvants for neuropathic and bone pain:				
Steroids	Dexamethasone, PO, IV	Adjuvant analgesic for inflammatory and anti-oedema effect	PO or IV: 0.3–0.6 mg/kg up to 10 mg	Onset: ND Peak: ND Duration: ND
	Methylprednisolone, PO	Adjuvant analgesic for inflammatory and anti-oedema effect	PO: 16 mg	Onset: ND Peak: ND Duration: ND
	Prednisone, PO	Adjuvant analgesic for inflammatory and anti-oedema effect	PO: 40–60 mg	Onset: ND Peak: ND Duration: ND
	Magnesium Sulphate	NMDA receptor blocker	IV bolus of 1–3 g followed by an infusion of 10 g over 20 h	Onset: ND Peak: ND Duration: ND
Bone pain:				
See NSAIDs, Steroids				
Visceral pain:				
See Steroids				

Note: Consider dose adjustment depending on renal and hepatic function and chronic therapy. Do not administer higher doses of NSAIDs as they are not effective. Adjust opioid dose carefully, especially in opioid-naïve patients. Calculate the correct opioid dose in opioid-tolerant patients as described in the text. Consider a multidrug approach according to the specific type(s) of pain, especially for EOL patients with severe pain and eventually de-escalate doses according to pain intensity. Prescribe a scheduled pain drugs administration with rescue doses. IM: intramuscularly; IN: intranasally; IV intravenous; ND: No Data available; NMDA: N-Methyl-D-aspartic acid; PO: orally administered.

## Data Availability

Data resulting from database search are available on request to the corresponding author.

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
