# Peer review of "Pain Management at the End of Life in the Emergency Department: A Narrative Review of the Literature and a Practical Clinical Approach"

_jcm, 2023, doi:10.3390/jcm12134357_

Round 1

Reviewer 1 Report

The review seems to be Sound, carefully Designed and is will written.

Conclusions are backed by results of Review.

the Main Pointe that should be adressed is that this is a review. In some Parts the manuscript Looks lobe a Meta Analysis. This should be clearly adressed.

my main concern is that only Nine of over 580 studies are included in the end. That means aproximately 99% if all primarily included studies are being exclused. This should be adressed More precisely.

Author Response

The review seems to be Sound, carefully Designed and is will written.

Conclusions are backed by results of Review.

  • Reply: We wish to deply thank the reviwer for its comments and appreciation of the paper.

the Main Pointe that should be adressed is that this is a review. In some Parts the manuscript Looks lobe a Meta Analysis. This should be clearly adressed.

  • Reply: Thank you for this comment. The main aim of this manuscript was to conduct a review of all available literature on the topic and to provide a comprehensive narrative review on the better, global approach to the management of End Of Life Care. So, we modified the abstract including the following sentences: “We provide a narrative summary of the findings and a review of the management of EOL pain in clinical practice, including i) the identification of the EOL patients and unmet palliative care needs, ii) a multidimensional, patient-centered assessment of the type and severity of pain, iii) a multidisciplinary approach to the management of end-of-life pain, including an overview of non-pharmacological and pharmacological techniques; and iv) the management of special situations, including rapid acute deterioration of chronic pain, breakthrough pain, and sedative palliation.” Moreover, the aim of the study has been modified as follows: “The aim of this manuscript was to i) identify the available literature on the management of severe EOL pain in the emergency department (ED), and iii) provide a clinical review on the identification of patients whorty of palliative care and on the multidimensional approach to the identification and management of pain in EOL patients in the ED , taking into account the available literature and the experience of the SedoAnalgesia Group (SAU) of the Società Italiana di Medicina d'Emergenza-Urgenza (SIMEU).”

my main concern is that only Nine of over 580 studies are included in the end. That means aproximately 99% if all primarily included studies are being exclused. This should be adressed More precisely.

  • Reply: Thank you again for this comment. Unfortunately, the vast majority of studies have nothing to do with the topic of interest. However, we agree with you and should explain the reasons for exclusion in more detail. In line with the previous response and to better clarify the process of our study selection, we have clearly indicated the search strategy for each database in Appendix A, re-run the search strategy, and corrected the total number of papers available. We have then provided a more detailed explanation of the reasons for exclusion for each database in Figure 1. Moreover, the followig sentences are reported in the discussion section: “As shown in Figure 1, despite the rigorous search strategy, most studies were excluded because they addressed topics other than end-of-life pain management, and only a few other studies included EOL patients. Most of these studies illustrated the causes of ED visits for EOL patients, the need for palliative care, factors influencing the possibility of palliative care pathways, and their global impact in the ED, but without data on pain management in EOL patients.”

Reviewer 2 Report

This paper reviewed the literatures on the management of severe end-of-life pain in the emergency department (ED) and summarized the medications for the management of end-of-life pain. The paper was very well-written. Reviews about the management of severe end-of-life pain in the ED is needed to guide current clinical care. I have a few minor comments to the authors.

1. Figure 1 needs some edition.

2. 3.2.3 Included studies’ results: it appears that the description in this part is lacking in sufficient details.

3. If possible, the authors may consider including a discussion on future research concerning the management of end-of-life pain in the emergency department.

Author Response

Reviewer 2:

This paper reviewed the literatures on the management of severe end-of-life pain in the emergency department (ED) and summarized the medications for the management of end-of-life pain. The paper was very well-written. Reviews about the management of severe end-of-life pain in the ED is needed to guide current clinical care. I have a few minor comments to the authors.

Reply: We would thank the reviewer for your insightful comment on our paper. We think this topic need to be addressed in more datail, to provide the best supportive care in all patients admitted to our emergency departments, despite his/her medical condition and prognosis.

  1. Figure 1 needs some edition.
  • Reply: We agree with you. We had some problem in the layout of the figure that we did not expect. Accordingly, the figure has been corrected. Moreover, we modified the figure adding the number and the reason of studies exclusion for each database.

  1. 2.3 Included studies’ results: it appears that the description in this part is lacking in sufficient details.
  • Reply: We agree with the reviewer. This paragraph has been modified as follows: “None of the included studies reported data on the effectiveness of different pain assessment strategies or on the prevalence of different types of pain; none of the included studies reported a comparison of different treatments in terms of effectiveness and safety. Bell et al. [29] reported on the percentage of medications used to treat pain and concluded that it is important to correctly identify EOL patients and initiate appropriate targeted treatment. Chan et al. [30] demonstrated the benefit of EOL care in the ED by increasing the percentage of symptomatic treatments administered without altering the dying process. Coyne et al. [31] reported the percentage of medications used and demonstrated the association between severe pain and the need for hospitalisation, especially in patients requiring opioids and with poor functional status. De Oliveira et al. [32] reported that palliative care needs were not met in 78.4% of patients, and only 56% of patients received medication for pain relief, showing that little attention was paid to palliative care compared to curative care. Rojas et al. [35] demonstrated the benefits of a palliative care protocol by reporting letters of appreciation from relatives.”

  1. If possible, the authors may consider including a discussion on future research concerning the management of end-of-life pain in the emergency department.
  • Reply: Thank you for this precious suggestion. As emergend in this SR, there is a lot of studies needed in this particular area. To suggest further areas of reaserch and to not make the manuscript redundant and repetitive, we modifed the conclusion as follows: “Identification of patients eligible for palliative and hospice care, identification of unmet needs for palliative care, patient-centred multidimensional pain assessment, and a multidisciplinary approach are critical steps in the care of patients in the ED. Further research is needed in these areas to provide the best evidence-based supportive therapy”
